# COMPLEXITY-BASED PROMPTING FOR MULTI-STEP REASONING

**Yao Fu**♠,*  **Hao Peng**♣, **Ashish Sabharwal**♣, **Peter Clark**♣, **Tushar Khot**♣
♠University of Edinburgh        ♣Allen Institute for AI
yao.fu@ed.ac.uk, haop@allenai.org, ashishs@allenai.org, peterc@allenai.org, tushark@allenai.org

## ABSTRACT

We study the task of prompting large-scale language models to perform multi-step reasoning. Existing work shows that when prompted with a chain of thoughts (CoT), sequences of short sentences describing intermediate reasoning steps towards a final answer, large language models can generate new reasoning chains and predict answers for new inputs. A central question is which reasoning examples make the most effective prompts. In this work, we propose complexity-based prompting, a simple and effective example selection scheme for multi-step reasoning. We show that prompts with higher *reasoning complexity*, i.e., chains with more reasoning steps, achieve substantially better performance on multi-step reasoning tasks over strong baselines. We further extend our complexity-based criteria from prompting (selecting inputs) to decoding (selecting outputs), where we sample multiple reasoning chains from the model, then choose the majority of generated answers from complex reasoning chains (over simple chains). When used to prompt GPT-3 and Codex, our approach substantially improves multi-step reasoning accuracy and achieves new state-of-the-art (SOTA) performance on three math benchmarks (GSM8K, MultiArith, and MathQA) and two BigBenchHard tasks (Date Understanding and Penguins), with an average +5.3 and up to +18 accuracy improvements. Compared with existing example selection schemes like manual tuning or retrieval-based selection, selection based on reasoning complexity is intuitive, easy to implement, and annotation-efficient. Further results demonstrate the robustness of performance gains from complex prompts under format perturbation and distribution shift.

## 1 INTRODUCTION

We consider the problem of prompting large language models for multi-step reasoning. Recent breakthroughs (Wei et al., 2022b; Wang et al., 2022b) show that language models, when large enough (>100B parameters), exhibit the emergent ability (Wei et al., 2022a) of performing complex multi-step reasoning when provided with only a few reasoning examples. In the regime of large models, prompting achieves comparable or even better performance than full training set finetuning while being substantially more sample-efficient (Wei et al., 2022b; Kojima et al., 2022; Lewkowycz et al., 2022). In particular, Wei et al. (2022b) show that chain-of-thoughts (CoT) prompts, sequences of short sentences describing intermediate reasoning steps towards final answers (Fig. 1A), can elicit strong reasoning capabilities from large language models for complex tasks such as math problems.

This work studies *example selection* in chain-of-thoughts multi-step reasoning. Example selection is a central problem in the prompting literature (Liu et al., 2022; Rubin et al., 2022; Su et al., 2022; Lazaridou et al., 2022). It asks what instances make the best prompts for solving the tasks of interest. For CoT prompting, example selection is further related to annotation efficiency, as CoT requires manually-annotated reasoning chains. For datasets where reasoning annotations are easy to obtain, one may want to know which annotated chains make the best prompt; if the annotations are hard to obtain, one may identify the best cases to annotate, rather than annotating the entire dataset.

---

*Work done during internship at Allen Institute for AI, code at `https://github.com/FranxYao/Complexity-Based-Prompting`

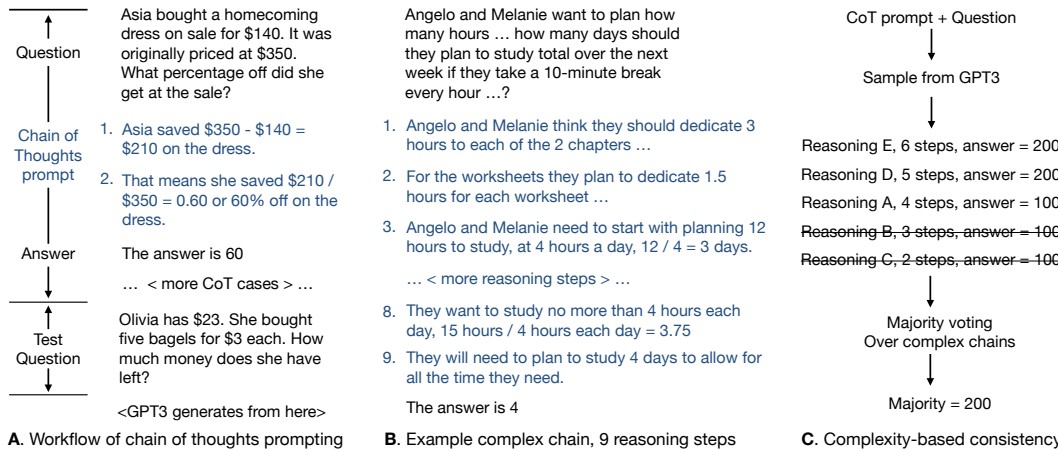

**A**. Workflow of chain of thoughts prompting   **B**. Example complex chain, 9 reasoning steps   **C**. Complexity-based consistency

Figure 1: **A**: Chain of thoughts (in blue) are intermediate reasoning steps towards a final answer. The input of CoT prompting is a stack of few (often 8) CoT cases before a test question. Then the language model will continue generating an output CoT for the test question. **B**: Chains of harder *reasoning complexity* are chains with *more reasoning steps* (9 steps in this case, v.s. only 2 steps in subfigure A). **C**: During decoding, we sample $N$ reasoning chains from the language model ($N = 5$ here), and take the majority answer over the $K$ ($K = 3$ here) most complex generated chains.

We propose *complexity-based prompting*, a new example selection scheme for chain-of-thoughts multi-step reasoning. Existing sample selection methods are usually based on manual tries (Wei et al., 2022b), heuristic rules (Wallace et al., 2019), optimization and search (Shin et al., 2020), or retrieval from a large training set (Rubin et al., 2022). Different from these schemes, complexity-based prompting chooses examples with complex reasoning chains, i.e., chains with more reasoning steps, as the prompt. Fig. 1A shows a simple example with 2 reasoning steps, versus the example in subfigure B is a complex case with 9 reasoning steps. As we will show in the experiments (§4.2), the reasoning performance of GPT-3 175B (Brown et al., 2020) clearly improves with the increased input prompt complexity, where complex prompts achieve better performance than simple prompts.

We further extend the complexity-based selection criteria from the input space (the prompts) to the output space (reasoning chains generated by the language model). Our extension is based on the idea of self-consistency (Wang et al., 2022b;a), where they sample multiple reasoning chains (instead of using greedy decoding) from the model that lead to possibly different answers, then choose the majority of the generated answers. Here we propose *complexity-based consistency*, where instead of taking a majority vote among all generated chains, we *vote over the top $K$ complex chains*, as shown in Fig. 1C. In §4.2, we will show that complexity-based consistency leads to further performance gains, on top of the existing gain from complexity-based prompting.

Putting everything together, our methods achieve new state of the art performance on three math benchmarks (GSM8K, MultiArith, and MathQA) and two BigBenchHard tasks (Date Understanding and Penguins) with substantial performance gains over Wei et al. (2022b). We show that, compared with existing sample selection schemes, complexity-based prompting achieves better performance in most cases (see §4.2). Furthermore, performance gains from complex samples are consistent in different prompt distributions (in-distribution, transfer, and noisily-labeled, see §4.2) and are also consistent with regard to alternative proxies for complexity (e.g., question or formula lengths, see §4.3) when the dataset does not contain annotated reasoning chains. A careful analysis shows that the number of reasoning steps is the most prominent factor, over confounders like prompt lengths or the number of input cases (§4.3). We hope this work will open new research possibilities in in-context learning, large language models, and multi-step reasoning.

## 2   RELATED WORK

**Emergent Abilities and Multi-Step Reasoning**      With the recent trend in scaling language models (Brown et al., 2020; Chowdhery et al., 2022), a central question is what *unique* abilities

emerge as models become large (Kaplan et al., 2020; Wei et al., 2022a). Generally, the ability to follow the format of given prompts (typically few-shot) thus solving the corresponding tasks (also referred as in-context learning), is something that large language models are particularly skilled at (Shin et al., 2020; Liu et al., 2021). Among the wide language understanding task spectrum, we are particularly interested in multi-step reasoning because of its two uniqueness: (1). multi-step reasoning is a task where large models substantially outperform smaller models (Wei et al., 2022b), versus performance gains on tasks like sentiment classification can be very limited with large models (Shin et al., 2020); (2). multi-step reasoning is where few-shot prompting starts to outperform full training set fine-tuning, even when fine-tuning is conducted on the same large model (Lewkowycz et al., 2022). This work takes an important step forward in multi-step reasoning by showing the critical role of prompt complexity.

**Chain-of-Thoughts Reasoning**     A prominent work demonstrating the multi-step reasoning of language models is chain-of-thoughts prompting (Fig. 1A), proposed by Wei et al. (2022b). They show that the reasoning ability can *only* be elicited by chain of thoughts, but not standard prompting where an answer directly follows a question without intermediate reasoning steps. Further works show that CoT can be improved by self-consistency (Wang et al., 2022b), pretraining the model with latex-formated data (Lewkowycz et al., 2022), context selection (Creswell et al., 2022), or even adding certain magic phrases like "Let's think step by step" (Kojima et al., 2022). The original CoT paper (Wei et al., 2022b) uses 8 manually written examples as the prompt, which are reused by most follow-up works. Our work sits in the context of CoT reasoning, and propose a new complexity-based prompt selection that substantially outperforms the original CoT.

**Example Selection for Prompting**     Designing prompts can be challenging due to the instability, as multiple works have shown the performance is sensitive to prompt, task, dataset, and model changes (Zhao et al., 2021; Lu et al., 2022; Su et al., 2022). Despite works on automatic prompt searching (which is more suitable for smaller models, e.g., Shin et al., 2020; Li & Liang, 2021), currently, prompt engineering for large models is (still) a community-wide collective trial and error effort (there is even a prompt marketplace named PromptBase). The difficulty is that *it is extremely hard to extract generalizable regularity from empirical observations that can form effective selection criteria*. One notable exception is similarity-based prompt selection, which retrieves the most similar training instances as the prompt for a given test case (Rubin et al., 2022). Yet for CoT prompting, retrieving different prompts for different test cases requires reasoning chain annotations for the whole training set, which compromises the advantage of being few-shot. Given this background, our core contribution is identifying complexity as an effective and robust selection criterion and in many cases, it outperforms existing prompt selection schemes while being annotation-efficient.

**Relation to Classical Semantic Parsing**     The procedure of chain of thoughts prompting is conceptually similar to classical semantic parsing where one generates a logical form then executes it upon a knowledge base to reach a final answer (Liang, 2016; Cheng et al., 2019). The practice of sampling then voting is also similar to marginalizing out semantic parses (Yin et al., 2018). There are further works linking the relationship between in-context learning and classical Bayesian inference (Wei et al., 2021; Xie et al., 2022). From our perspective, we tend to view chain-of-thoughts as flexible, language model styled "logical forms" which are "executed" by the language model itself. We leave further study on connecting classical parsing and CoT to future work.

## 3 COMPLEXITY-BASED PROMPTING

We study multi-step reasoning tasks, and use math word problems, mathematical problems expressed in natural language, as our testbed. This task, as is measured by solve rate (accuracy), is to predict the answer (typically a number) of a given math word problem via intermediate steps. We follow the chain-of-thoughts prompting framework and compare all prompting schemes using GPT-3 `text-davinci-002` and Codex `code-davinci-002`. An example problem, as well as the chain-of-thoughts workflow, is shown in Fig. 1A. The input is a stack of a few (often 8) CoT cases followed by a test question, then the language model continues generating an output CoT for the test question. Our goal is to improve the reasoning accuracy by identifying and exploiting more effective input and output reasoning chains.

## 3.1 SELECTING COMPLEX SAMPLES AS PROMPTS

Our method is to simply choose complex prompts over simple ones. We hypothesize that language models' reasoning performance will increase if we use complex instances as in-context "training example," as they intuitively subsume simpler instances (Richardson & Sabharwal, 2022). We define complex instances as instances with more reasoning steps (Fig. 1B), as the name "multi-step reasoning" indicates. Note that using reasoning steps as the notion of complexity is also the practice of previous works like (Sugawara et al., 2018; Lai et al., 2021). We further define a step as a line, separated by the linebreak "\n".

There are two aspects that need more discussion: (1) The notion of complexity. There are other complexity indicators than number of steps, such as questions lengths or the length of the underlying formula for solving a given problem. We will show that the trend that better performance comes with more complex prompts is *consistent across various complexity indicators, such as question lengths and formula lengths*. Consequently, for datasets that do not have annotated reasoning chains, we can use questions lengths to identify complex instances, then only annotate the identified few-shot instances, thus reducing the annotation cost. (2) Confounders of number of steps. The increase in performance with more complex examples in the prompt could be explained by correlated factors like the increase in the total number of reasoning steps in the prompts or just the increased length of the prompt. To account for this, we evaluate prompts with simpler examples but the same number of reasoning steps (e.g. 24 cases with 3 steps vs. 8 cases with 9 steps, both of 72 steps in total). We also consider prompts of the longest lengths (but not most steps). We show that *the number of steps per example* is the most prominent source of performance gains over confounders.

## 3.2 COMPLEXITY-BASED CONSISTENCY

Complexity-based prompting can be further enhanced with a new output selection method following the same intuition, which we present in this section. Existing evidence shows that the expressive neural models can take *shortcuts* during reasoning, relying on spurious correlations that inevitably exist in the training data (Mudrakarta et al., 2018; Sugawara et al., 2018; Lai et al., 2021). This often leads to suboptimal generalization to unseen data. To alleviate this issue, we explicitly promote outputs with more complex reasoning chains at inference time. Specifically, our method follows the self-consistency practice in Wang et al. (2022b), which samples $N$ reasoning chains for a test question. Different reasoning chains may lead to different answers, and Wang et al. (2022b) takes the majority answer as the prediction. In our case, instead of voting among all $N$ chains, we only vote among top $K$ ($K \leq N$) complex (more steps) reasoning chains, as shown in Fig. 1C. We dub our method *Complexity-based Consistency*. Note that when $K = N$ we recover the original self-consistency method. In our experiments, we set $N$ to 50, and observe that the optimal $K$ is always smaller than $N$ (typically 30-40). This provides clear evidence that voting among more complex reasoning chains generalizes better than voting among all. We also show that if we do the opposite and vote among answers produced by $K$ simplest reasoning chains, the accuracy is always worse than voting among all. This further validates that complex chains, not simple chains, should be considered more during decoding.

## 4 EXPERIMENTS

We first discuss our experimental settings in §4.1. In §4.2 and §4.3, we present the following results: (1) our method substantially outperforms the original CoT (Wei et al., 2022b). It establishes new state-of-the-art results on three math reasoning datasets (GSM8K; Cobbe et al., 2021; MultiArith; Roy & Roth, 2015; MathQA; Amini et al., 2019), a temporal reasoning task (Date Understanding; Suzgun et al., 2022), and the referential game task (Penguins; Suzgun et al., 2022). On StrategyQA (Geva et al., 2021), a commonsense reasoning dataset, our approach matches the existing state-of-the-art performance. (2) Performance gains from complex prompts are consistent: no matter what large model we use (GPT-3 or Codex), what distribution the prompt come from (in-distribution, noisy distribution, and distribution shift), or whether there exists prompt format perturbation or confounders, complex prompts consistently outperform simpler prompts; (3) Compared with other example selection schemes (random, heuristic and retrieval), complexity-based example selection often achieves the best or competitive results with minimal annotation budget. In

the appendix, we discuss further experimental results, including a performance gain breakdown, results on smaller models, output step distribution, and further experiments on voting

## 4.1 EXPERIMENTAL SETTINGS

**Datasets** We use three math word problems datasets (GSM8K, MultiArith, and MathQA) and three non-math reasoning (StrategyQA, Date Understanding, and Penguins) as our testbed. We choose GSM8K and MultiArith also because they are the datasets used by prior work on CoTs (Wei et al., 2022b; Wang et al., 2022b; Kojima et al., 2022), allowing fair comparison to existing methods. MathQA's annotation are much noisier than others, and we use it to evaluate the robustness of our approach. There are 1.3K test instances in GSM8K, 600 in MultiArith, and 600 in MathQA. For each dataset, we randomly draw 200 instances from the training data to create a validation split. The cost of prompting GPT-3 is proportional to the size of test set. For the non-math datasets, StrategyQA is a multi-step commonsense reasoning task with 800 test instances. Date Understanding is a temporal reasoning task with 250 test instances. Penguins is a referential game (a referential game asks questions referring to different objects, e.g., is penguin A older than penguin B and C) with 146 test instances. Both Date Understanding and Penguins are subsets of the BigBench Hard datasets (datasets that previously fine-tuning struggles with, see Suzgun et al., 2022). Evaluating on a 200-instances validation set costs about 6-8 US dolars for greedy decoding (1 output chain) and $12-$24 for sampling 50 output chains. Prompting Codex is currently (November 2022) free and we hope OpenAI could continue making it free to the community.

**Language Models** We consider two paradigms: fine-tuning and prompting. For fine-tuning, we report the existing SOTA performance: a fine-tuned GPT3 with a verifier (Cobbe et al., 2021) on GSM8K, a relevance and LCA operation classifier (Roy & Roth, 2015) on MultiArith and a customized sequence to sequence model (Amini et al., 2019) on MathQA. For prompting, we consider the following language models: (1). LaMDA (Thoppilan et al., 2022), a 137B model used as the baseline in Wei et al. (2022b); (2). PaLM (Chowdhery et al., 2022), the primary 540B model used in the CoT papers; (3). Minerva (Lewkowycz et al., 2022), a 540B large model that trains on LATEXdata; it achieves SOTA performance in math reasoning on GSM8K; (4). GPT-3 175B (text-davinci-002 from Brown et al., 2020) (5). Codex (code-davinci-002 from Chen et al., 2021, also 175B). We further consider the DiVeRSe (Li et al., 2022) method which equips an additional trained verified to GPT-3/ Codex and is the previous SOTA on GSM8K. Our experiments are mostly conducted on GPT-3 and Codex because they are the accessible to the public thus more reproducable. LaMDA, PaLM and Minerva are not accessible to the public, and their numbers are from their corresponding papers.

**Prompts and Hyperparameters** The training sets of GSM8K and MathQA contain human annotated reasoning chains, within which we search for complex prompts. MultiArith does not have annotated reasoning chains, so we consider two strategies. (1). *in-distribution annotation*, which uses question lengths as an alternative proxy for complexity, then manually annotates reasoning chains for complex questions; (2). *prompts transfer* from GSM8K training data. All prompts for math datasets contain 8 cases (a case = a question + a chain of thoughts + an answer). For non-math datasets, since they do not have annotated reasoning chain, we again, use question length as the complexity proxy and manually annotates reasoning chains for complex questions. Following Kojima et al. (2022), we add "Let's think step by step" before the reasoning chains for all prompting schemes to improve the performance.

## 4.2 MAIN RESULTS

**Overall Test Performance on Math Datasets** Table 1 shows the overall performance of models. We consider two decoding strategies: (1) greedy decoding (the first block of Table 1) and (2) majority vote (§3.2; the second block of Table 1). Note that PaLM and Minerva are more than three times larger than GPT-3 and Codex, the model we use to evaluate our method, and Minerva is additionally pretrained on latex data. Therefore, they are by no means comparable to the methods based on GPT-3 or Codex. We nevertheless outperform all of them.

We consider three prompting schemes: (1). *Handcrafted CoT* constructed originally by Wei et al. (2022b) then reused in following-up works (Wang et al., 2022b; Kojima et al., 2022; Wang et al., 2022a). (2). *Random CoT*: randomly drawing samples from the training set. GSM8K and MathQA

Table 1: Complexity-based prompting, when applied on Codex (code-davinci-002), achieves new state-of-the-art performance on GSM8K, MultiArith, and MathQA. † models are not publicly accessible, and the numbers are from their papers. Our performance gain (**+blue**) is computed over the original handcrafted CoT used in Wei et al. (2022b), which is our primary baseline. Our methods substantially increase the performance over Wei et al. (2022b), with an average **+5.3** gain on GPT-3 and **+6.2** on Codex.

| | | #Params | GSM8K | MultiArith | MathQA |
|---|---|---|---|---|---|
| Previous finetuning SOTA | | $\leq$175B | 57.0 | 60.5 | 37.4 |
| **Greedy decoding** (Wei et al., 2022b) | | | | | |
| LaMDA† (Thoppilan et al., 2022) | | 137B | 17.1 | 51.8 | - |
| PaLM† (Chowdhery et al., 2022) | | 540B | 58.1 | 94.7 | - |
| Minerva† (Lewkowycz et al., 2022) | | 540B | 58.8 | - | - |
| Text-davinci-002 | Handcrafted CoT | 175B | 48.1 | 90.8 | 30.1 |
| | Random CoT | 175B | 49.7 | 89.5 | 34.8 |
| | **Complex CoT** | 175B | **55.4 (+7.3)** | **94.2 (+3.4)** | **36.0 (+5.9)** |
| Code-davinci-002 | Handcrafted CoT | 175B | 61.0 | 95.8 | 29.3 |
| | Random CoT | 175B | 60.4 | 97.3 | 40.5 |
| | **Complex CoT** | 175B | **66.6 (+5.6)** | **95.8 (+0.0)** | **47.3 (+18.0)** |
| **Voting among multiple outputs** (Wang et al., 2022b) | | | | | |
| LaMDA† (Thoppilan et al., 2022) | | 137B | 27.7 | 75.7 | - |
| DiVeRSe (Li et al., 2022) | | 175B | 82.3 | 99.8 | - |
| PaLM† (Chowdhery et al., 2022) | | 540B | 74.4 | 99.3 | - |
| Minerva† (Lewkowycz et al., 2022) | | 540B | 78.5 | - | - |
| Text-davinci-002 | Handcrafted CoT | 175B | 64.0 | 98.2 | 43.8 |
| | Random CoT | 175B | 62.0 | 95.2 | 48.5 |
| | **Complex CoT** | 175B | 71.5 | 97.3 | 49.5 |
| | **+ Vote Complex** | 175B | **72.6 (+8.6)** | **98.7 (+0.5)** | **50.2 (+6.4)** |
| Code-davinci-002 | Handcrafted CoT | 175B | 74.6 | 99.7 | 55.0 |
| | Random CoT | 175B | 77.3 | 99.3 | 58.2 |
| | Complex CoT | 175B | 82.6 | 99.7 | 58.6 |
| | **+ Vote Complex** | 175B | **82.9 (+8.3)** | **99.8 (+0.1)** | **60.0(+5.0)** |

training data have reasoning chain annotations, so we directly use them. MultiArith does not have reasoning annotations, so we randomly sample eight training cases then annotate the chains manually. (3). *Complex CoT*. For GSM8K and MathQA, we choose eight training cases with the most numbers of reasoning steps; For MultiArith, we use the question length as the proxy for complexity, and manually annotate reasoning chains for the eight training cases with the longest questions. Complex prompt selection results in substantially more reasoning steps: it averages 9.0 steps on GSM8K, while the handcrafted and random schemes yield 3.4 and 2.8 steps respectively. The trends are similar on the other two datasets. The handcrafted prompts uses the same fixed prompt for all three datasets but the cases within the prompt does not come from any of the datasets (so they are in a sense, out of distribution). Complex prompts and random prompts all come from their corresponding training sets (so these two are in a sense, in-distribution).

As Table 1 shows, our method achieves substantially better performance than the baselines. Besides, our proposal of voting among complex chains outperforms voting among all (last two lines in Table 1. Furthermore, our performance using GPT-3 is close to PaLM and Minerva, two language models that are more than three times larger than GPT-3 and are not publicly accessible. These results directly demonstrate the effectiveness of our methods.

**Consistent Performance Improvements on Different Reasoning Tasks** Table 2 shows that the advantage of complex prompts holds for different types of reasoning tasks. When prompted with complex examples, GPT-3/ Codex achieves new SOTA performance on Date Understanding and Penguins datasets where complex prompts consistently improves performance over simpler prompts.

Table 2: Complex prompts give comparable performance to PaLM on StrategyQA (commonsense reasoning), and achieve new state of the art performance on Date Understanding (temporal reasoning), and Penguins (referential game) datasets. Accuracy gain (**+blue**) is computed over the original handcrafted CoT used in Wei et al. (2022b;a). All results use greedy decoding.

|  | **Prompt** | **#Params** | **StrategyQA** | **Date Understanding** | **Penguins** |
|---|---|---|---|---|---|
| PaLM | Handcrafted | 540B | 77.8 | 79.2 | 65.1 |
| Text-davinci-002 | Handcrafted | 175B | 66.9 | 82.8 | 76.7 |
|  | Simple | 175B | 71.1 | 76.4 | 61.0 |
|  | Complex | 175B | **77.0 (+10.1)** | 82.4 (-0.4) | 79.5 (+2.8) |
| Code-davinci-002 | Handcrafted | 175B | 73.1 | 86.0 | 78.1 |
|  | Simple | 175B | 74.4 | 83.2 | 69.8 |
|  | Complex | 175B | 73.9 (+0.8) | **86.8 (+3.6)** | **80.8 (+2.7)** |

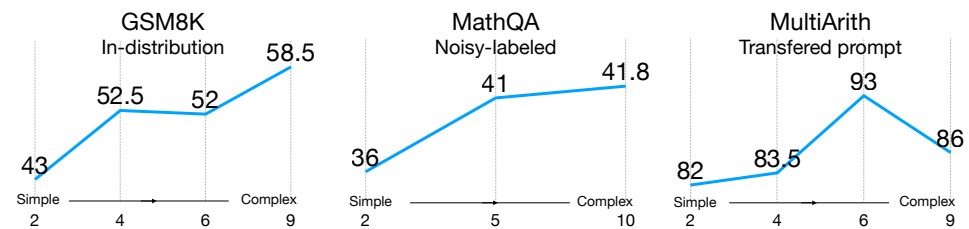

Figure 2: Validation set performance. X-axis means reasoning steps and y-axis means accuracy. More reasoning steps in prompts overall achieve higher accuracy when prompts are in-distribution (left), noisily labeled (middle), and out of distribution (right).

**Consistent Performance Improvements in Different Prompt Distributions**    We investigate the performance of our complexity-based prompting when the prompts are: (1) from clean in-distribution training set (GSM8K); (2) from noisy annotation (MathQA); (3) are transferred from another dataset (MultiArith). Here as MultiArith does not have annotated reasoning chains, and their questions are similar to the ones in GSM8K; we use (transfer) prompts from GSM8K for MultiArith. Figure 2 shows that in general, more complex prompts achieve better performance, and this trend is consistent in all the three settings, except for one particular case on MultiArith.

**Comparison to other Example Selection Schemes**    As we view the reasoning complexity as the basis of a new example selection scheme, we compare it with existing selection schemes. We consider: (1) *random* selection; (2) *Centroid*, where we select examples whose question embeddings (produced by a pretrained sentence encoder Reimers & Gurevych, 2019) are the closest to the embeddings of all other questions, i.e., questions at the center part of the dataset. The intuition is that centroid examples may be the most typical or representative cases of a dataset; (3) *Retrieval*, where we retrieve questions from a training set whose embeddings are closest test question measured in Euclidean distance. Notably, there are important differences between retrieval and other methods: retrieval uses different prompts for different test cases, while other methods use fixed prompts for all. Therefore, the annotation cost of retrieval scales with the size of the test set, and is usually about the full-training-set-sized annotation (more than 10K cases), while others only require few-shot annotation (in our cases, only 8 examples).

As shown in Table 3, complexity-based selection outperforms all other methods on GSM8K and MultiArith. On MathQA, although retrieval-based selection outperforms complexity-based selection, it has two importance restrictions that we do not have: (1) as mentioned, retrieval requires substantially more CoT annotation, while we only requires few-shot; (2) the performance of retrieval is critically determined by how similar the test cases and the training questions are to each other, and the similarity may not always hold. We further note that on MathQA, many dev. questions are quite similar to their retrieved training questions (some of them only have minor changes like "8 apples plus 9 bananas" to "10 apples plus 5 bananas" while the underlying computations are the same). So in general, complexity-based prompting has the advantage of good performance while being annotation efficient.

Table 3: Comparison to other prompt example selection schemes (validation accuracy). On GSM8K and MultiArith, complexity-based selection outperforms all the baselines. On MathQA, although retrieval performs better than complexity, it requires substantially more annotation.

| | #Annotations | GSM8K | MultiArith | MathQA |
|---|---|---|---|---|
| Random | Few-shot (8) | 52.5 | 86.5 | 33.0 |
| Centroid | Few-shot (8) | 52.0 | 92.0 | 32.0 |
| Retrieval | Full training set ($\geq 10000$) | 56.0 | 88.0 | **69.5** |
| Complexity (ours) | Few-shot (8) | **58.5** | **93.0** | 42.5 |

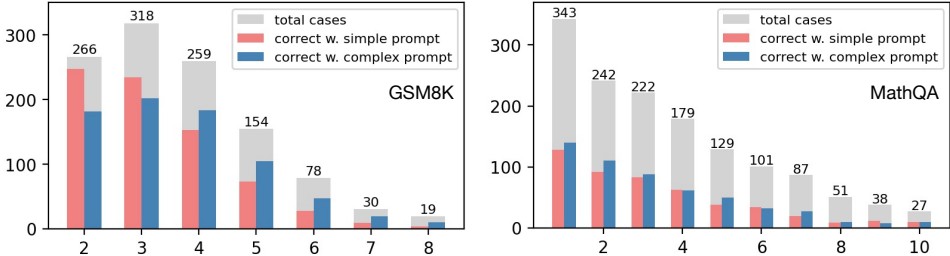

Figure 3: X-axis means reasoning steps of dev set cases and y-axis frequency. The direction of generalization on the two datasets is intriguing and show different patterns: on GSM8K, simple prompts perform better for simple cases ($\leq 3$ steps) while complex prompts perform better for complex cases; on MathQA, simple prompts do not have advantages for simple case and complex prompts seem to perform better on most of the groups.

Table 4: Alternative complexity measure: Q Len. = question length, F Len. = formula length. More complex prompts consistently outperform simpler ones.

| | Q Len. | GSM8K | F Len. | MathQA |
|---|---|---|---|---|
| Simple | 70 | 49.0 | 7.5 | 37.5 |
| Mid | 226 | 51.0 | 55 | 33.5 |
| Complex | 815 | **52.5** | 165 | **43.5** |

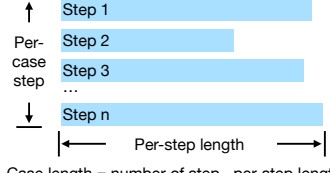

Figure 4: Relationship between confounders.

**Direction of Generalization** Intuitively, one may attribute the improvements of complexity-based prompting to accuracy gains on complex test cases. Yet interestingly, our analysis suggests the opposite. Fig. 3 compares the validation set accuracy of complex and simple prompts, varying the number of reasoning steps in the *gold annotation*. We observe a clear trend on both GSM8K and MathQA: complex prompts perform on par with simple prompts on *hard* cases, while achieving more clear gains on cases with fewer number of reasoning steps. This finding suggests that complexity-based prompting generalizes to simpler test cases. We conjecture that this is because the reasoning capabilities elicited by complex prompts may cover simple questions better. Further investigation into the underlying mechanism is definitely interesting, and is left to future work.

## 4.3 ANALYSIS

In this section, we develop in-depth analysis supporting our claims. All experiments in this section are performed on validation sets. We first show that the performance improvements with more reasoning complexity is consistent in terms of: (1). different proxies for complexity and (2). different step formatting. Then we show that the number of reasoning step is the most prominent factor for performance improvements over its confounders. Finally, we strengthen our conclusion of complexity-based consistency, and show that the optimal performance is always achieved by majority voting over complex chains, not simple chains.

Table 5: Confounder analysis. Although there exist confounders like number of cases or total prompt length, the number of reasoning step is the most prominent factor for performance gain given moderate per-step length.

| *More number of simple cases* v.s. *less but complex cases* | GSM8K | | MathQA | |
|---|---|---|---|---|
| Total reasoning step | 72 | 72 | 45 | 45 |
| Number of cases in prompt | 24 | 8 | 19 | 8 |
| Per-case reasoning step | 3 | 9 | 2.25 | 5.625 |
| Accuracy | 51 | **58.5** | 37.5 | **42.5** |
| *Most number of reasoning steps* v.s. *longest prompt* | GSM8K | | MathQA | |
| Number of cases in prompt | 8 | 8 | 8 | 8 |
| Length of prompt | 12.6k | 8.4k | 7.6k | 4.9k |
| Number of total reasoning step | 59 | 72 | 32 | 45 |
| Per-step length | 112 | 74 | 137 | 52 |
| Accuracy | 57 | **58.5** | 31 | **42.5** |
| *Shorter per-step length* v.s. *Longer per-step length* | GSM8K | | MathQA | |
| Number of total reasoning step | 72 | 72 | 45 | 45 |
| Number of cases in prompt | 8 | 8 | 8 | 8 |
| Per-step length | 36 | 74 | 37 | 52 |
| Accuracy | 51.0 | **58.5** | 30.5 | **42.5** |

**Alternative Proxies for Complexity**     Complexity-based prompting is equally applicable when the data does not come with reasoning chain annotations, as we have already shown that selecting cases with longest questions also improves performance (§4.2). In Table 4, we confirm that in addition to number of steps, either using questions length or formula length as the measure of complexity, the optimal performance is achieved with complex prompts. These results mean that the effectiveness of complex prompts are consistent with regard to the notion of complexity.

**Confounder Analysis**     All experiments so far keeps the number of instance to be 8 in all prompts. Yet when choosing complex examples with more reasoning steps, we observe that the following factors are correlated (also illustrated in Fig. 4): (1). when *per-case reasoning step* increases (for example, in GSM8K we choose cases with 9 reasoning steps), the *total number of step* in the whole prompt also increase (in GSM8K, we have 8 cases in the prompt, so there are 8 × 9 = 72 steps in total). This might be compensated by using *more number of simple cases* (e.g., 24 simple cases, each with 3 steps, can also make 72 steps in total). These factors are shown in the upper part of Table 5. (2). when *per-case step* increases, the *full length of the prompt* (= number of characters) also increases, which may be compensated by longer (more characters) but less step examples. These factors are shown in the lower part of Table 5. From the accuracy results we can see that: (1). keeping full number of reasoning steps the same, using more number of simple cases does not outperform less number of complex cases; (2). longest prompts does not outperform complex prompts. (3). yet we do need a moderate per-step length because keeping total number of step 72, moderate per-step length prompts outperforms shorter per-step length prompts. This means that despite the existence of confounders, the number of reasoning steps per example is the most prominent factor for performance gain given moderate per-step length.

## 5   CONCLUSION

This paper proposes a new complexity-based instance selection scheme for prompting language models to perform multi-step reasoning. In addition to substantial performance improvements on math word reasoning tasks, our methods exhibit multiple advantages such as being intuitive, annotation-efficient, and robustly effective in different in-context learning settings. We hope this work will open new research possibilities in prompting, language models, and multi-step reasoning.

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

Table 6: GSM8K validation set performance improvements broken down on to various design choices. More than half of the accuracy improvements can be attributed to our complexity-based prompting and output selection (indicated by †).

| Greedy Decoding | Acc. | Majority Vote | Acc. |
|---|---|---|---|
| CoT Original | 43.5 | CoT Original | 55.5 |
| Add "Let's think step by step" (Kojima et al., 2022) | 48.5 (+5.0) | Add "Let's think step by step" and change "Q: " to "Question:" | 61.0 (+5.5) |
| Use complex prompt† | 54.0 (+5.5) | Use complex prompt† | 67.0 (+6.0) |
| Change "Q: " to "Question: " | 58.0 (+4.0) | Voting within complex sample† | 71.0 (+4.0) |

Table 7: Complexity-based prompting is an emergent ability of large models. If applied to smaller models, complex prompts cannot induce significant performance gain (recall in Table 1 complex prompts induces average +6.2, maximum +18.0 accuracy gain on Codex).

|  | MultiArith | | GSM8K |
|---|---|---|---|
|  | text-curie-001 6.7B | Flan-T5 11B | Flan-T5 11B |
| Handcrafted | 3.8 | 51.5 | 19.5 |
| Random | 2.0 | 53.3 | 21.0 |
| Complex | 3.3 | 51.3 | 21.0 |
| Δ | -0.5 | -0.2 | +1.5 |

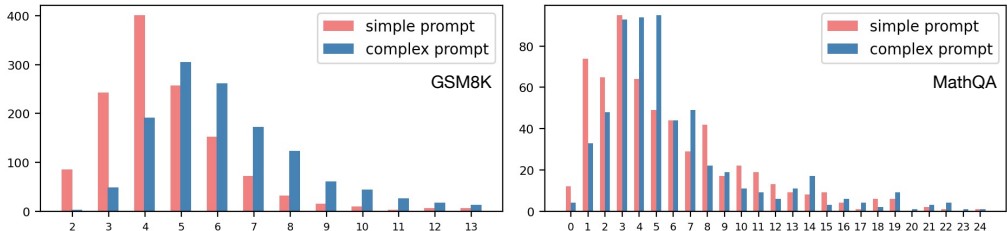

Figure 5: Output step distribution. X-axis means reasoning steps and y-axis means frequency. As a sanity check, complex prompts indeed induce complex outputs than simple prompts.

## A    APPENDIX

Below we list some additional experimental results.

**Performance Improvements Breakdown**    Improving prompting performance commonly requires prompt engineering. A common scepticism or criticism towards performance improvements on leaderboards like Table 1 is whether the accuracy gains are from the proposed method or other independent engineering efforts. Here we list all the efforts we made for leaderboard climbing on Table 6. While techniques like adding "Let's think step by step" (Kojima et al., 2022) improves the accuracy, the performance gains can be primarily attributed to complexity-based prompting, validating the effectiveness of our methods.

**Performance on Small Models**    Does smaller models also enjoy the performance gain from complex prompts? Unfortunately, this seems to be not the case. As is shown in Table 7, complex prompts cannot induce meaningful performance gain over the original or random prompts. This indicates that, like the chain-of-thoughts prompting itself, *complexity-based prompting is also an emergent ability* that exist only when the model scale is large enough.

**Output Step Distribution**    As a sanity check, in Fig. 5, we show that complex prompts induce complex reasoning than simple prompts (Codex outputs on GSM8K and MathQA). This means that complex prompts are indeed discouraging the model from taking easier reasoning path, thus potentially avoiding shortcuts.

Table 8: Sensitivity analysis on step formatting. Complex prompts consistently lead to better performance with regard to different step formatting.

|  | Linebreak "\n" | Period "." | Explicit "step i" | Semicolon ";" |
|---|---|---|---|---|
| GSM8K-Complex | **58.5** | 54.5 | 52.0 | 54.0 |
| GSM8K-Simple | 43.0 | 40.5 | 42.0 | 41.0 |
| MathQA-Complex | **42.5** | 39.0 | 36.0 | 39.5 |
| MathQA-Simple | 34.0 | 34.5 | 33.5 | 37.0 |

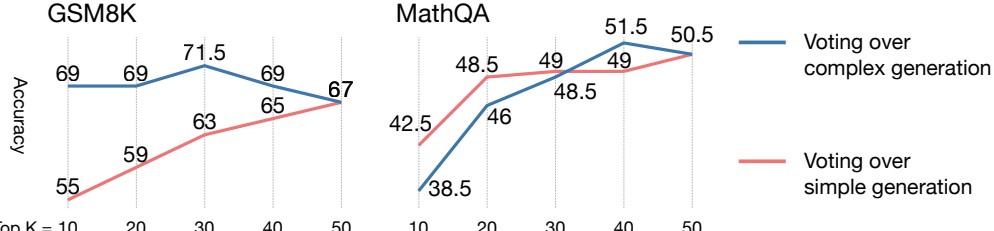

Figure 6: Majority voting over top K complex / simple generated samples. The optimal performance is achieved on selecting complex samples over simple samples.

**Sensitivity Analysis on Step Format** A common concern with prompting is that the performance can be sensitive to the format of the input (Shin et al., 2020; Liu et al., 2022) and may change with input perturbations. Here we study one important perturbation: the splitter of steps, which is an existing concern of CoT-styled prompting in Rong; Akyurek & Akyurek (2022). As alternatives to the linebreak "\n" we use, we consider two more types of splitters: (1). explicit phrases "step i" (2). two punctuation marks, period "." and semicolon ";" The performance is shown in Table 8. Although these perturbations do have an influence on the performance, complex prompts consistently lead to better performance with regard to different step formatting.

**Voting among Complex Chains Outperforms Voting among All** Now we analyze the properties of complexity-based consistency, which generalizes the reasoning complexity selection criteria from the input space (prompts) to the output space (sampled solutions from the language model). Complexity-based consistency first sample $N$ reasoning chains from the model, then take the majority answer voted from the top $K$ complex chains. Here we set $N = 50$, and control $K = 10, 20, 30, 40, 50$. Note that when $K = 50$ we recover the original self-consistency (no complexity-based selection). As a further comparison, we consider the other way around: instead of voting over top $K$ complex samples, we vote over top $K$ simple samples. As is shown in Fig. 6, we see: (1). voting over simple samples always *underperform* full sample; , indicating this is not a correct direction for performance; (2). both datasets achieve the best performance on some $K^\star < N$ with complex voting. These results again, validate the choice of complex samples.

