# OpenReview forum: "Complexity-Based Prompting for Multi-step Reasoning"
_ICLR.cc/2023/Conference — ICLR 2023 poster_

### Official Review · Reviewer_QWaf · 2022-10-21

**Confidence:** 3
**Correctness:** 3
**Technical Novelty And Significance:** 3
**Empirical Novelty And Significance:** Not applicable
**Recommendation:** 8

**Clarity, Quality, Novelty And Reproducibility:**

This work is clearly written and the message is evident to the reader without any struggle. The experiments are sound and the quality of the work is well done.
They propose a new complexity-based instance selection scheme which is shown to be effective in multistep reasoning. This is a valid and useful idea and it contributes to the field of research.

**Strength And Weaknesses:**

- The idea is nice, focusing on the instance selection problem by looking at the complexity is interesting.

- The authors define a step as a line, separated by the linebreak “\n”. They do look at the length of the input at some point but I am wondering if there are some side effects from defining a step like this. Do the authors know whether processing steps together would provide additional context or is it too difficult to learn from?

- From the results (e.g., Table 1) it looks like the number of parameters is an important factor in the overall performance. Have the authors looked into the effectiveness of their proposed approach with language models of different sizes?

- Typo: section 3.1: the n. umber of

**Summary Of The Paper:**

This paper investigates the example selection problem which is selecting reasoning examples that make the most effective prompts when prompting large language models with a chain of thoughts (CoT) prompt. The authors apply their approach at both prompting and decoding time. Evaluating on multi-step reasoning task, they observe improvements over the baselines.

**Summary Of The Review:**

This work is well-motivated, clearly explained, and sufficiently supported by analysis. There are some questions for the authors (see section Strength And Weaknesses), however overall it's a valuable contribution to the field.

---

> ### Author Response · Authors · 2022-11-17
> **Response to reviewer QWaf**
>
> We thank the reviewer for the detailed review and finding our work effective and clear. Below we address the reviewer’s comments:
>
> ## Detailed discussion about step definition
>
> > “The authors define a step as a line, separated by the linebreak “\n”. They do look at the length of the input at some point but I am wondering if there are some side effects from defining a step like this.”
>
> Formally, we tend to view a step as a sentence expressing one computation/ equation in natural language, and we view the linebreak “\n” as the separator of steps. There could be other separators, like a dot “.” in which case all steps become one line, but the number of step does not change.
>
> The reason we use linebreak as the separator (thus one step is one line) is because it performs very well (better than other separators, see Table 7 in the paper), and we think using linebreak is indeed an important yet untold trick in the prompt engineering literature. We guess probably this is because linebreak is also a very common separator in the training set. Otherwise, we have not yet observed any other side effects that could potentially damage the performance.
>
> > “Do the authors know whether processing steps together would provide additional context or is it too difficult to learn from?”
>
> We have tried if we replace the linebreak “\n” to a period “.” (thus putting the steps together, if that is what “processing steps together” refers to in the question). Yet in this way, we still think there are the same number of steps (just different separators). We have also tried other separator “;”, or letting the model explicitly say “step 1”, “step 2”. Among all the separators, linebreak gives the best performance (See table 7 in the paper). So we do believe that other separators make it harder for the model to learn reasoning. This behavior is intriguing and we do not have deeper explanations other than linebreak being a frequent separator in the pre-training data. We definitely would like to explore more about the underlying mechanism in the future work.
>
> ## Additional experiments from more language models and new SOTA results with Codex
>
> > “it looks like the number of parameters is an important factor in the overall performance. Have the authors looked into the effectiveness of their proposed approach with language models of different sizes?”
>
>
> Yes indeed. For chain-of-thought to work, the model scale is crucial. This is because chain-of-thought reasoning is an emergent ability of language models (Wei et. al. 2022) – abilities that only exist when the model is sufficiently large. Since our complexity based prompting is built upon chain-of-thought, we also require the backbone model being sufficiently large. In the updated paper Table 5, we show that when using two smaller models (GPT-3 curie-davinci-001 6.7B and Flan-T5 11B), different prompts do not have significantly different performance.
>
> That being said, we do verify that our method is effective for other large models than GPT-3. Specifically, we show when applying our complexity-based prompting to Codex (175B), we achieve new state-of-the-art performance on all three math reasoning tasks we consider. Complex prompts bring an average of +6.2 accuracy gain and maximum +18.0 accuracy gain over the original handcrafted prompt used in Wei et. al. 2022a. We further show the applicability is not just on math reasoning: complex prompts also achieve SOTA results on two BigBench Hard tasks (Date Understanding for temporal reasoning and Penguins for referential game). We refer the reviewer to the updated paper Table 1 and  2 for the full results. Below we give a quick Preview:
>
> |  | GSM8K | MultiArith | MathQA |
> | - | ---- | ---- | ---- |
> | Previous SOTA |  82.3 (Li et. al. 2022) | 99.8 (Li et. al. 2022) |  37.4 (Amini et. al. 2019) |
> | Codex w. Handcrafted CoT  | 73.6 | 99.7 | 55.0 |
> | Codex w. Complex CoT (ours) | 82.9 | 99.8 | 60.0 |
>
> |  | StrategyQA | Date Understanding | Penguins |
> | - | ---- | ---- | ---- |
> | Previous SOTA | 77.8 (wei et. al. 2022a) | 79.2 (Suzgun et. al. 2022) | 65.1  (Suzgun et. al. 2022) |
> | Ours Complexity-based Prompting | 77.0 | 86.8 | 80.8 |
>
>
> Also we have fixed the typo pointed out by the reviewer (thanks!)
>
>
> ## References
> Wei et. al. 2022a. Emergent Abilities in Large Language Models.
>
> Li et. al. 2022. On the Advance of Making Language Models Better Reasoners
>
> Suzgun et. al. 2022. Challenging BIG-Bench tasks and whether chain-of-thought can solve them.
>
> Amini et. al. 2019. MathQA: Towards Interpretable Math Word Problem Solving with Operation-Based Formalisms

---

### Official Review · Reviewer_Gum9 · 2022-10-25

**Confidence:** 3
**Correctness:** 3
**Technical Novelty And Significance:** 3
**Empirical Novelty And Significance:** 3
**Recommendation:** 6

**Clarity, Quality, Novelty And Reproducibility:**

The paper is well written and easy to follow. The proposed in the paper is simple but it seems it improved the existing results.


**Strength And Weaknesses:**

Strength:
1- The paper considered example selection for a multi-step reasoning task and it picked examples with complex reasoning chains, i.e., chains with more reasoning steps, as the prompt.

2- In comparison to existing example selection schemes like manual tuning or retrieval-based selection, complexity-based prompting sounds intuitive, easy to implement.

3- The paper also proposed complexity-based consistency, where instead of taking a majority vote over all generated chains, we vote over the top K complex chains. Their experimental results also show that voting among more complex reasoning chains generalizes better than voting among all.


Weakness:
1- The paper mentioned that majority voting over top K complex generated samples.performs better than the top-k simple generated samples. However the results in Figure 5 for the MathQA are different. Is there any explanation for that?

2- The generalization results are not clear.  According to Figure 3 complex prompts improve the performance on simple cases, however the accuracy of the complex prompts is very low. I think some statistics on the dataset where how many examples are there in each bucket may make it more clear. For example in GSM8k, for how many samples the number of reasonings are 8, etc.

3- I understand the paper focused on generalizability based on the number of steps. However, I am wondering if the accuracy error is because of the ask complexity or the complexity of the numbers (length of the digits)


**Summary Of The Paper:**

The paper proposed complexity-based prompting for multi-step reasoning and it showed prompts with more reasoning steps, achieving substantially better performance on math word reasoning tasks over strong baselines. Their experimental results show that such sample selection and voting over top-k complex output samples improve the accuracy.

**Summary Of The Review:**

The paper proposed a simple example selection that improves the existing results. The paper also mentioned that rather than majority voting over the output sample voting over top-k complex outputcsamples (those with more number of steps) contributes to the accuracy improvement. The paper still needs to clarify/justify the results in Figure 5 since it conflicts with these statements.

---

> ### Author Response · Authors · 2022-11-17
> **Response to reviewer Gum9 (2/2)**
>
> ## Updating the generalization results
> > “According to Figure 3 complex prompts improve the performance on simple cases, however the accuracy of the complex prompts is very low. I think some statistics on the dataset where how many examples are there in each bucket may make it more clear.”
>
> We thank the reviewer’s suggestion. We have updated the dev set from 200 cases to 1000 cases and annotated the number of cases for each bucket. After expanding the dev set size, the observation is indeed different than previous 200-sized dev set:
> * In MathQA, complex prompts outperform simple prompts for questions of 1-6 steps (each bucket more than 100 cases), but the two prompts perform similar for questions of 8 - 10 steps (100 cases in total)
> * In GSM8K, on simple cases (2 and 3 steps, each more than 200 cases), **simple prompts perform better than complex prompts**. This is an intriguing observation and it means complex prompts are not uniformly better than simple prompts. Complex prompts are better than mid-level and complex cases (more than 3 steps).
>
> At this stage, the observation of the generalization direction is indeed intriguing, but still inconclusive. We would like to reiterate that our main contribution is showing the effectiveness and robustness of complex prompts, which we do have more exciting results showing that complexity-based prompting + Codex lead to SOTA performance on all 3 math datasets we consider and 2 more multi-step reasoning datasets (see above responses to all reviewers, also updated paper Table 1 and 2). Yet we do think the direction of generalization is interesting, and apparently different from traditional fine-tuning. We definitely will be investigating more about generalization in our future work.
>
> ## Clarifying other potential confounders
> > “I am wondering if the accuracy error is because of the ask complexity or the complexity of the numbers (length of the digits)”
>
> We are not sure what the reviewer means by "ask complexity". Is it a typo of “task complexity” or does it mean “question complexity”?
> * If it means task complexity, then we are still unsure about what task complexity refers to because both simple and complex prompts are applied to the same task.
> * If it means question complexity, we do think question complexity, or more precisely question length, is also a factor related to number of reasoning steps (the more complex a question is, the more steps it requires). In table 6, we have shown that one can also use question length as the proxy of complexity, in which prompts with complex/ long questions still output prompts with simple/ short questions.
>
> About the complexity of the numbers/ length of the digits, we would like to reassure that this factor is constant for both simple and complex prompts: all numbers in simple/ complex prompts and test instances are less than three digits. Additionally, even if the digits change, Aman et. al. 2022 have shown that the actual number does not influence the performance, and it is the reasoning that play a more important role – so the length of digits should not be an important factor.
>
> ## References
> Madaan and Yazdanbakhsh. 2022. Text and Patterns: for Effective Chain of Thought it Takes Two to Tango.

---

> > ### Comment · Reviewer_Gum9 · 2022-12-03
> > **Thanks for the clarification.**
> >
> > I would like to thank authors for the clarification.
> > I am willing to increase the score since the new results and explanations were convincing.

---

> ### Author Response · Authors · 2022-11-17
> **Response to reviewer Gum9 (1/2)**
>
> We thank the reviewer for the detailed comments. We understand the reviewer’s concerns, which are also very important to us. Below we clarify:
>
> ## Clarifying the results in Figure 5
>
> > “The paper mentioned that majority voting over top K complex generated samples.performs better than the top-k simple generated samples. However the results in Figure 5 for the MathQA are different.”
>
> K is a hyperparameter. During testing, we only need **the best K** for the best performance, and it is OK if other Ks are not as good. So in Figure 5 (after paper update it now becomes Figure 6), the point is that we do not need **all** K in the complex to outperform all K in the simple. We only need **one** K in the complex to perform the best. This is to say, in the figure, we do not need the blue curve to always be above the red curve. We only need the peak of the blue curve to be above the red curve.
>
> In practice, our workflow is that we first use the dev set to observe that the optimal performance is achieved by voting among top 40 complex samples over the full 50 samples (which gives 51.5% accuracy and better than other Ks in either complex or test sample). So we fix K = 40 and use complex voting for the test set (which consequently gives us SOTA performance on the datasets we consider).
>
> ## Reiterating the core contribution of our work
>
> In addition to the discussions around complex voting, we would like to raise the reviewer’s attention more to the following advantages of our method, which we believe have a bigger impact beyond decoding strategy:
> * New SOTA results by applying complexity-based prompting to Codex. As is suggested by reviewerJ9JD, we have added experiments applying our method to another language model backbone (Codex). The results are great: we achieve new state-of-the-art performance on all three math reasoning tasks we consider. Complex prompts brings an average of +6.2 accuracy gain and maximum +18.0 accuracy gain over the original handcrafted prompt used in Wei et. al. 2022a. See updated paper Table 1 for details.
> * Applicability to more reasoning tasks. Per reviewer ZKDK’s suggestion, we have added experiments applying our method to a commonsense reasoning task (StrategyQA), a temporal reasoning task (Date Understanding), and a referential game task (Penguins). Our methods achieve new SOTA results on two of them, further validating our methods applicability.  See updated paper Table 2 for details.
> * Robustness of our method: complexity-based prompting is a very reliable principle of choosing prompts, applicable to different large models, no matter how complexity is defined, no matter if the prompt distribution is shifted, no matter if the task is math or other multi-step reasoning, no matter if there exist format perturbation or confounders.  We are not aware if there exist any other prompting methods that have this level of reliability and robustness.
>
> Combining these three major advantages, we do believe that this work would benefit the community of prompting and language model research.

---

### Official Review · Reviewer_ZkDK · 2022-10-25

**Confidence:** 5
**Correctness:** 3
**Technical Novelty And Significance:** 2
**Empirical Novelty And Significance:** 2
**Recommendation:** 3

**Clarity, Quality, Novelty And Reproducibility:**


Concerns on clarity:

- The definition of complexity is not clear, and it is not detailed how the complexity is calculated. What is considered a “reasoning step”.
- Adding "Let's think step by step" is a confounding factor that may confound the main results. It would be better to report results without this factor. (table 2)

Concerns on novelty:

- The method attempts at finding a way to design a good chain-of-thought, which is a valid task.
- The proposed method is based on the intuition that prompts with higher reasoning complexity, i.e., chains with more reasoning steps, would achieve better performance. This can be seen as a special case of heuristic-based methods. It is not clear what is fundamentally novel about this work
- It is not clear what would be the cause of this effect shown in the paper. Longer reasoning chains might be breaking down the problems into smaller bite-sized problems that LLMs can handle. It would be interesting to see more analysis on this. Perhaps the reasoning chain length is the wrong notion, but the “granularity” of the reasoning steps is the right metric.


Concerns on reproducibility:

- The notion of complexity is not consistent across datasets and tasks, and needs hand annotation. This limits the applicability of the proposed method.
- Some steps makes this paper a bit difficult to reproduce, like how to annotate the reasoning chain for MultiArith.
- It is not clear if this effect can be reproduced on other datasets.

**Strength And Weaknesses:**

Strength:
- The proposed method is intuitive and easy to implement, as claimed by the authors.
- The paper is well-written and easy to follow. The authors provide detailed experimental results and analysis.
- The proposed method is a simple and effective example selection scheme for multi-step reasoning, which is an improvement over existing methods that require hand-engineering.

Weaknesses:
- The proposed method is only evaluated on GPT-3, and it would be interesting to see if the proposed method also works on smaller language models.
- The proposed method is only evaluated on math word reasoning tasks. It would be interesting to see if the proposed method also works on other tasks, such as the Big-Bench tasks.

**Summary Of The Paper:**

This paper studies the task of prompting large-scale language models to perform multi-step reasoning, and proposes a simple and effective example selection scheme for multi-step reasoning based on reasoning complexity. The proposed method is to select prompts with higher reasoning complexity, according to the author's definition of complexity. The proposed method is evaluated on math word reasoning tasks, and is shown to outperform baseline CoT construction methods. The method is also robust to format perturbation and distribution shift.


**Summary Of The Review:**

In this work, the authors study the task of prompting large-scale language models to perform multi-step reasoning, and propose a simple and effective example selection scheme for multi-step reasoning based on reasoning complexity. The proposed method is evaluated on math word reasoning tasks, and is shown to outperform baseline CoT construction methods.  There are some concerns about the novelty of this work. In addition, the proposed method is only evaluated on GPT-3 and a few reasoning tasks, and it would be interesting to see if the proposed method also works on smaller language models and more tasks.

---

> ### Author Response · Authors · 2022-11-18
> **Response to reviewer ZKDK (4/4)**
>
> ## Clarification about different notions of complexity and reproducibility
> > “The notion of complexity is not consistent across datasets and tasks, and needs hand annotation. This limits the applicability of the proposed method.”
>
> We beg to differ: we choose to use different complexity nations, aiming to promote the applicability of our methods in various settings. Some datasets (like GSM8K) come with annotated reasoning chains, for which we can directly use the number of reasoning steps to select the prompt. Some datasets do not have annotated reasoning chains (like MultiArith), so one inevitably needs other complexity metrics to identify complex examples, then annotate manually (also this annotation is quite easy and it takes less than 5 minutes for all experiments in our paper). Our practice of using multiple notions of complexity does not limit the applicability, but on the contrary, expands the applicability of our method to different datasets with different notions of complexity (recall that we have already demonstrated that our method is effective in three more datasets). Our key advantage is that no matter what complexity notion (#steps, question length, etc.) we used, complex prompts consistently perform better than simpler ones.
>
> > “Some steps makes this paper a bit difficult to reproduce, like how to annotate the reasoning chain for MultiArith.”
>
> Annotating the reasoning chain for the prompt examples is the requirement that all chain-of-thought based papers share, and all of them do this painlessly because it is just easy and intuitive (e.g., Chung et. al. 2022 annotated 57 new tasks without trouble) . In our case, to annotate the reasoning chain for MultiArith, just write down the reasoning in plain language, step by step. Below is an example:
>
>     Question: A teacher had 29 worksheets to grade. If she graded 25, but then another 29 were turned in, how many worksheets would she have to grade?
>
>     Reasoning:
>     There are 29 initially <- step 1
>     After grading 25 of them, there are 29 - 25 = 4 worksheets <- step 2
>     Adding the new 29 worksheets makes 4 + 29 = 33 worksheets <- step 3
>     The answer is 33 <- answer
>
> Annotating our 8-instances prompts is very easy and takes less than 5 minutes. We will also release the code, the model output, and all prompts used.
>
>
> > “It is not clear if this effect can be reproduced on other datasets.”
>
> In the revised paper, we have 6 datasets in total, and all of them show complex prompts clearly outperforms simple prompts (see updated paper, Table 1 and 2). Practically, to reproduce the effect on more datasets, first identify simple/ complex cases using the length of question as the complexity notion, then write down the reasoning in plain language, step by step. Here we show an example from Date Understanding
>
>     Question: Tomorrow is 11/12/2019. What is the date 10 days ago in MM/DD/YYYY?
>     Reasoning:
>     If tomorrow is 11/12/2019, then today is 11/11/2019. <- step 1
>     This means 10 days ago is 11/01/2019. <- step 2
>
> Again, in our practice, annotating 8-instances prompts for StrategyQA/ Date Understanding/ Penguins takes less than 5 minutes.
>
> ## Discussions about the underlying mechanism
> > “It is not clear what would be the cause of this effect shown in the paper.”
>
> Yes indeed. The underlying mechanism of complexity-based prompting, and broadly emergent abilities of large language models is intriguing. We are currently actively looking into the causes and would definitely take this as steps of future research.
>
> ## References
>
> Wei et. al. 2022a. Chain of Thought Prompting Elicits Reasoning in Large Language Models.
>
> Wei et. al. 2022b. Emergent Abilities in Large Language Models.
>
> Suzgun et. al. 2022. Challenging BIG-Bench tasks and whether chain-of-thought can solve them.
>
> Wolfson et. al. 2020. Break It Down: A Question Understanding Benchmark
>
> Amini et. al. 2019. MathQA: Towards Interpretable Math Word Problem Solving with Operation-Based Formalisms
>
> Yang et. al. 2018. HotpotQA: A Dataset for Diverse, Explainable Multi-hop Question Answering
>
> Dua et. al. 2019. DROP: A Reading Comprehension Benchmark Requiring Discrete Reasoning Over Paragraphs
>
> Lai et. al. 2021. Why Machine Reading Comprehension Models Learn Shortcuts?
>
> Wang et. al. 2022. Self-Consistency Improves Chain of Thought Reasoning in Language Models
>
> Zhou et. al. 2022. Least-to-Most Prompting Enables Complex Reasoning in Large Language Models
>
> Sun et. al. 2022. Recitation-Augmented Language Models

---

> ### Author Response · Authors · 2022-11-18
> **Response to reviewer ZKDK (3/4)**
>
> ## Clarification about novelty and key advantages of complexity over other prompting methods
>
> > “chains with more reasoning steps, would achieve better performance. This can be seen as a special case of heuristic-based methods. It is not clear what is fundamentally novel about this work”
>
> The key differences between complexity-based prompting and the community’s mainstream prompt writing practice (which is mostly trial-and-error) is that complexity-based prompting is a very reliable principle of choosing prompts, applicable to different large models, no matter how complexity is defined, no matter if the prompt distribution is shifted, no matter if the task is math or other multi-step reasoning, no matter if there exist format perturbation or confounders. We are not aware if there exist any other prompting methods that have this level of reliability and robustness.
>
> We would be very careful and not take the inspiration of complex prompts for granted. In all of the subsequent work on CoT (Wang et. al. 2022. Zhou et. al. 2022. Sun et. al. 2022. Creswell et. al. 2022. Kojima et. al. 2022. inter alia) , none of them considered using complex prompts. An important reason why it is not obvious is that complex prompts depart from standard machine learning setting (where training and testing data follow the same distribution) and intentionally introduces a distribution shift between training and evaluation – it selects training instances that are substantially more complex than the test cases in terms of the number of reasoning steps. The fact that complex prompts achieve SOTA performance is also unconventional, as distribution shift usually induces performance drop in traditional fine-tuning (Lake and Baroni. 2018. Dollak et. al. 2018. Liu et. al. 2022. Inter alia), yet we get performance increases. All these distributional generalization properties are important yet unsolved in both fine-tuning and prompting literature, and we do believe our work provides novel and effective methods with new generalization insights given the background.
>
> ## Clarification about granularity
> > “Perhaps the reasoning chain length is the wrong notion, but the “granularity” of the reasoning steps is the right metric.”
>
> We definitely agree that the reasoning granularity is an important dimension to look upon (which we do have related experiments about per-step length in Table 8). Our concern here is that there are not yet well-established metrics that give a clear definition and quantification of granularity. In comparison, the number of reasoning steps is easy to measure (one just counts), and is well-established by the community (e.g., the following papers also use the number steps: Yang et. al. 2018, Dua et. al. 2019, Wolfson et. al. 2020, Lai et. al. 2021, Wei et. al. 2022,  inter alia).. It is also quite effective in our experiments as we see increasing number of reasoning steps clearly increase the performance.
>
> That being said, we do have experiments that study factors potentially related to granularity, which is the per-step length (average number of characters of each step) in the paper figure 4 and table 8. The answer is that we do need a moderate per-step length prompt: prompts where each step has 112 characters or 36 characters cannot outperform our complex prompt where each step has 84 characters. This means that given the number of step being large, one step can neither be too short nor too long (it should have a moderate level of granularity).

---

> ### Author Response · Authors · 2022-11-18
> **Response to reviewer ZKDK (2/4)**
>
> ## Clarification on definitions of complexity
> > “The definition of complexity is not clear, and it is not detailed how the complexity is calculated. What is considered a “reasoning step”.”
>
> The quick answer is that the definition of the reasoning step depends on the data. E.g. one computation in GSM8K or one subquestion about a factual information in a multi-hop QA dataset. It is usually very straightforward to define a step simply by looking at the data (which is a common practice of many previous works (see more examples below). But note that even if there is some ambiguity, our method does not rely on knowing the exact #reasoning steps for each problem -- just an approximate relative ordering is sufficient.
>
> As for example reasoning steps, for math word problems, usually, one step is about one computation expressed in a sentence:
>
>     Jorge spent 24 tickets * $7 per ticket = $168 total. <- step 1
>     After applying the discount, Jorge spent $168 * 0.50 = $84. <- step 2
>
>
> For date understanding, one step is usually about one sentence comparing two dates:
>
>     If tomorrow is 11/12/2019, then today is 11/11/2019. <- step 1
>     This means 10 days ago is 11/01/2019. <- step 2
>
> In the general multi-hop question answer literature, one step is usually one-sentence reasoning about one entity/ fact/ supporting evidence, depending on the dataset (Yang et. al. 2018, Dua et. al. 2019, Wolfson et. al. 2020, Lai et. al. 2021, Wei et. al. 2022,  inter alia). In our work, to compute the complexity (the number of step), we simply count the number of sentences.
>
> One important point here is that, **for all notions of complexity we considered (token length, #steps, question length), selecting more complex examples consistently helps**. As different notions of steps or complexity measure are usually correlated with each other, no matter what the exact definition of step we use, as long as we follow the complexity rule, we are very likely to get a better performance.
>
> ## Clarification about confounder
> > “Adding "Let's think step by step" is a confounding factor that may confound the main results. It would be better to report results without this factor.”
>
> We add “Let’s think step by step” to **all prompts** being compared in this paper, **not just our own**. For example, in Table 1, we have also added “Let’s think step by step” to the baseline original handcrafted CoT (thus improving the baseline performance). Since all prompts are taking the advantage of “Let’s think step by step”, this is a constant factor and a fair comparison.
>
> Yet we do have conducted additional experiments showing that removing “Let’s think step by step” from all prompts does not change the performance order, as is shown below:
>
> | | GSM8K | MathQA |
> | - | ---- | ---- |
> | GPT-3 Handcrafted Prompt, no “step by step” | 48.5 | 33.5 |
> | GPT-3 Simple Prompt, no “step by step” | 34.0 |  33.0 |
> | GPT-3 Complex Prompt, no “step by step” | **55.5** |  **35.0** |
>
> The results show that complex prompt still perform better than the other two prompts.

---

> ### Author Response · Authors · 2022-11-18
> **Response to reviewer ZKDK (1/4)**
>
> We thank the reviewers’ detailed comments. We understand the reviewer’s concerns, which are also very important to us. Below we clarify:
>
> ## Results on other language models and new SOTA performance with Codex
> > “The proposed method is only evaluated on GPT-3, and it would be interesting to see if the proposed method also works on smaller language models.”
>
> We first note that the chain-of-thought reasoning ability itself is an emergent ability: abilities that only exist in sufficiently large language models (see Wei et. al. 2022ab, Suzgun 2022, also in our paper’s related work section). Complexity-based prompting is based on chain-of-thought, which means that it also requires the underlying language model being large enough. GPT3 and Codex are the only publicly available models of such sizes and we have experiments on both of them in the paper now. Also note that Codex is currently free upon access request. Keeping this in mind, we did run experiments on smaller models based on the reviewer’s suggestion. In our updated paper table 5, we test a smaller version of GPT-3 (text-curie-001 6.7B) and a instruction-tuned T5 (FLAN-T5 11B) on MultiArith and GSM8K, and the performance of different prompts are similar to each other (difference within 2 points accuracy).
>
> Yet this does not imply that our method cannot be applied to other large language models. As suggested by reviewer D9JD, we further conduct experiments on Codex (175B). Using complexity-based prompting, we achieve new state-of-the-art performance on all three math datasets we consider with an average +6.2, up to +18.0 accuracy gain over the original handcrafted CoT from Wei et. al. 2022a. The results in updated in Table 1 in the paper, below we provide a quick preview:
>
> |  | GSM8K | MultiArith | MathQA |
> | - | ---- | ---- | ---- |
> | Previous SOTA |  82.3 (Li et. al. 2022) | 99.8 (Li et. al. 2022) |  37.4 (Amini et. al. 2019) |
> | Codex w. Handcrafted CoT  | 73.6 | 99.7 | 55.0 |
> | Codex w. Complex CoT (ours) | 82.9 | 99.8 | 60.0 |
>
> ## New SOTA performance on two Big-Bench Hard tasks
> > “It would be interesting to see if the proposed method also works on other tasks, such as the Big-Bench tasks.”
>
> We thank the reviewer for this suggestion, which we believe helps strengthen our results. Per the reviewer’s request, we conduct additional experiments on two BigBench Hard (hardest subset of BigBench, see Suzgun et. al. 2022) tasks that require explicit multi-step reasoning: Date Understanding (a temporal reasoning dataset) and Penguins (a referential game dataset). Our complex prompts with Codex achieve new state of the art performance on these two tasks. We further consider a commonsense reasoning task, StrategyQA, and show that complex prompts with Codex achieves similar performance to the much larger PaLM 540B, the previous SOTA. The results are in the updated paper, Table 2. Below we provide a quick preview:
>
> |  | StrategyQA | Date Understanding | Penguins |
> | - | ---- | ---- | ---- |
> | Previous SOTA | 77.8 (wei et. al. 2022a) | 79.2 (Suzgun et. al. 2022) | 65.1  (Suzgun et. al. 2022) |
> | Ours Complexity-based Prompting | 77.0 | 86.8 | 80.8 |

---

### Official Review · Reviewer_D9Jd · 2022-11-01

**Confidence:** 4
**Correctness:** 4
**Technical Novelty And Significance:** 4
**Empirical Novelty And Significance:** 3
**Recommendation:** 8

**Clarity, Quality, Novelty And Reproducibility:**

The paper is nicely written and easy to understand. Experiments are complete (although lacking a bit on different LM backbones).

I think the method is novel for prompting, and the findings are interesting.

The code and data are not shared.


**Strength And Weaknesses:**

The proposed method is very simple and it shows consistent improvement over the previous methods. It substantially improves the GPT-3 performance on chain-of-thoughts reasoning. The ablation study shows the method is very robust across many different angles.

Weakness: The experiments are only carried out on GPT-3, and the performance is generally inferior to those of PaLM and Minerva. The authors probably cannot get the performance on that two models, it would be great if they can illustrate on other large models, like OPT and Codex.


**Summary Of The Paper:**

The paper proposes a simple complexity-based prompting method for chain-of-thought prompting. The idea is to prompt with more complex questions: The authors use example questions with more reasoning steps (9 per question). When combining the predictions from multiple prompts, they choose the top K predictions with the most reasoning steps. Results show consistent improvement on 3 maths reasoning datasets, across various complexity of questions and formats of the prompts, and perform better than other prompt example selection methods.

**Summary Of The Review:**

Questions:
1. How does changing the complexity in input change the complexity in output? I did not find an answer in the paper. Does it often result in more reasoning steps in output?
2. For the "retrieval"-based method in Table 3, how did you get all the annotations for the whole dataset? The CoT annotation is probably different from the original rationales in the dataset.
Typo: Page 2 second paragraph: "n. umber".

Overall, I think the paper is nicely written and good for ICLR.

---

> ### Author Response · Authors · 2022-11-17
> **Response to reviewer D9JD**
>
>
> We thank the reviewer for finding this work simple, effective and robust. Below we address the reviewers comments.
>
> ## New SOTA results by applying complexity-based prompting to Codex
>
> This is a fantastic suggestion! By the time we submitted the paper, we did not realize that Codex is better than GPT-3 (works showing its advantages are mostly after our submission, e.g., Suzgun et. al. 2022). With Codex, our approach establishes new SOTA performance on all three math datasets and two additional Bigbench Hard (hardest subset of bigbench, see Suzgun et. al. 2022) datasets that require explicit multi-step reasoning (Date Understanding for temporal reasoning and Penguins for referential game). On math datasets, complexity-based prompting leads to an average +6.2, up to +18.0 accuracy gain over the original handcrafted CoT from Wei et. al. 2022a. We have updated the new results in the paper, Table 1 and 2. A quick preview is below:
>
> |  | GSM8K | MultiArith | MathQA |
> | - | ---- | ---- | ---- |
> | Previous SOTA |  82.3 (Li et. al. 2022) | 99.8 (Li et. al. 2022) |  37.4 (Amini et. al. 2019) |
> | Codex w. Handcrafted CoT  | 73.6 | 99.7 | 55.0 |
> | Codex w. Complex CoT (ours) | 82.9 | 99.8 | 60.0 |
>
> |  | StrategyQA | Date Understanding | Penguins |
> | - | ---- | ---- | ---- |
> | Previous SOTA | 77.8 (wei et. al. 2022a) | 79.2 (Suzgun et. al. 2022) | 65.1  (Suzgun et. al. 2022) |
> | Ours Complexity-based Prompting | 77.0 | 86.8 | 80.8 |
>
> ## Other Important Comments and Questions
> > “How does changing the complexity in input change the complexity in output?”
>
> Complex prompts elicit complex outputs (outputs of more reasoning steps). We have updated the paper (Figure 5) with output step distribution. Here are some quick statistics:
> * On GSM8K, the average output step, simple v.s. complex = 4.97 v.s. 6.64
> * On MathQA, the average output step, simple v.s. complex = 5.85 v.s. 6.35
>
> > “For the "retrieval"-based method in Table 3, how did you get all the annotations for the whole dataset?”
>
> GSM8K and MathQA already have the annotation, so we just use them. MultiArith does not have the annotation, yet their questions are similar to GSM8K ones, so we retrieve from the GSM8K training set and use the retrieved GSM8K instances as prompts for MultiArith.
>
> Typo and Code:
> We have also fixed the typo and will release the code, data, and model outputs upon decision.
>
> ## References
> Suzgun et. al. 2022. Challenging BIG-Bench tasks and whether chain-of-thought can solve them.
>
> Wei et. al. 2022a. Chain of Thought Prompting Elicits Reasoning in Large Language Models.
>
> Amini et. al. 2019. MathQA: Towards Interpretable Math Word Problem Solving with Operation-Based Formalisms

---

### Author Response · Authors · 2022-11-17
**Response to all reviewers and summary of updates**

We thank the reviewers for their insightful comments. Below is a summary of new results and updates made according to the collective opinions from all reviewers.

(Disclaimer: to include the additional experiments that the reviewers suggested, the revised paper has 11 pages in total, which is one page more than the limit. We will update the paper to be within the page limit after rebuttal.)

## New SOTA results by applying complexity-based prompting to Codex
(per request from reviewer D9JD and ZkDK)

We thank the reviewers for asking more results of other language models than GPT-3. In the updated paper, Table 1, we show when applying our complexity-based prompting to Codex (175B), we achieve new state-of-the-art performance on all three math reasoning tasks we consider. Complex prompts bring an average of +6.2 accuracy gain and maximum +18.0 accuracy gain over the original handcrafted prompt used in Wei et. al. 2022a. A quick preview is below:

|   | gsm8k | MultiArith | MathQA |
| - | ---- | ---- | ---- |
| Previous SOTA | 82.3 (Li et. al. 2022) |  99.8 (Li et. al. 2022) | 37.4 (Amini et. al. 2019) |
| Codex w. Handcrafted CoT |  73.6 | 99.7 | 55.0|
| Codex w. Complex CoT (ours) | 82.9 | 99.8 | 60.0 |

We note there are three important factors that collectively lead to the SOTA performance:
1. The backbone language model Codex (Chen et. al. 2021) which contains strong potential for complex reasoning
2. Chain of thoughts prompting (Wei et. al. 2022) unlocks the reasoning ability of Codex to get good performance
3. Complexity based prompting (ours) substantially lifts the performance from high to SOTA.

## New SOTA results on BigBench Hard Subtasks
(per request from reviewer ZkDK)

We thank the reviewer for asking for the results of more datasets. In the updated paper, Table 2, we show complexity-based prompting achieves new state of the art performance on two bigbench hard (hardest subset of bigbench, see Suzgun et. al. 2022) datasets that require explicit multi-step reasoning (Date Understanding for temporal reasoning and Penguins for referential game). We also show that complexity-based prompting achieves comparable performance to existing SOTA on StrategyQA, an important commonsense reasoning benchmark. A quick preview is below:

| - | StrategyQA | Date Understanding | Penguins |
| - | ---- | ---- | ---- |
| Previous SOTA | 77.8 (wei et. al. 2022a) |  79.2 (Suzgun et. al. 2022) | 65.1  (Suzgun et. al. 2022) |
| Ours Complexity-based Prompting | 77.0 | 86.8 | 80.8 |

## Results on smaller models

(per request from reviewer ZkDK and QWaf)

We thank the reviewer for asking for the results on smaller models. In the updated paper, Table 5, we add results from two smaller models (GPT-3 text-curie-001 6.7B and Flan-T5 11B). In these smaller models, handcrafted prompts, random prompts, and complex prompts have very similar performance (performance delta within 2 points). This means that, like chain-of-thought prompting, the property of complexity-based prompting is also an **emergent ability** that only exists when the model is sufficiently large.

We further note that the fact that complexity-based prompting only works for large models should be viewed as an intrinsic behavior of scaling language models, rather than a disadvantage of our method. Similar behaviors that only exist in large models are also observed in recent/ concurrent work (Wei et. al. 2022a, b, Chung et. al. 2022, Suzgun et. al. 2022, inter alia). Currently, the study of emergent abilities is still in a very early stage, and we believe our work plays an important role in the bigger picture by showing that complexity-based prompting is a reliable rule-of-thumb that leads to SOTA performance.

## Other important concerns

There are certain concerns about the applicability (ZKDK), the reproducibility (ZKDK), and the effectiveness of sampling complex chains (Gum9), which we clarify in the corresponding responses. There are also questions regarding details about output step distribution (D9JD), confounders (ZKDK and Gum9), and generalization (Gum9), which we have addressed accordingly. Generally, reviewers agree that our work is “simple and effective”, “the message is evident to the reader without any struggle”, and “sufficiently supported by analysis”. We refer the readers to our individual responses to reviewers for more details. We will release code, data, and model outputs after the decision.

## References
Wei et. al. 2022a. Chain of Thought Prompting Elicits Reasoning in Large Language Models.

Wei et. al. 2022b. Emergent Abilities in Large Language Models.

Chung et. el. 2022. Scaling Instruction-Finetuned Language Models.

Suzgun et. al. 2022. Challenging BIG-Bench tasks and whether chain-of-thought can solve them.

Li et. al. 2022. On the Advance of Making Language Models Better Reasoners

Amini et. al. 2019. MathQA: Towards Interpretable Math Word Problem Solving with Operation-Based Formalisms

---

### Decision · Program_Chairs · 2023-01-20

**Decision:**

Accept: poster

**Justification For Why Not Higher Score:**

While this paper shows good results with the proposed method, the method itself is quite straightforward and the paper is largely empirical.

**Justification For Why Not Lower Score:**

The paper proposes a method that does work well and will be of interest to the increasingly growing community of researchers and practitioners working with large language models.

**Metareview: Summary, Strengths And Weaknesses:**

The paper presents a new approach to prompting for language models, called complexity-based prompting, which achieves state-of-the-art results on several math reasoning tasks. The approach uses a measure of complexity to select the most complex chain of thoughts from a set of generated chains, which are then used to prompt the language model. The authors show that these prompts result in a significant increase in performance compared to standard CoT prompting. Although the reviewers have initially raised concerns about the novelty of the method, and the fact that it was evaluated on one language model and only mathematical datasets, the authors were able to address these concerns in the discussion period, where they provided new results on the Codex language model and on two hard subtasks from the BigBench dataset, showing that their approach outperforms existing methods. They have also addressed concerns about the reproducibility, applicability, and effectiveness of their approach. Overall, the paper shows that complexity-based prompting is a promising approach for improving the performance of language models on complex reasoning tasks.

**Note From Pc:**

if the above contains the word "oral" or "spotlight" please see: "oral" presentation means -> notable-top-5% and "spotlight" means -> notable-top-25%. As stated in our emails, we are disassociating presentation type from AC recommendations

**Summary Of Ac-Reviewer Meeting:**

We did not meet because 3/4 reviewers agree that this paper should be published (8, 8, 6) and the only outlier reviewer who gave a low score (3) did not engage in the discussion period, although his main concerns of applying the method on other models and datasets were successfully addressed by the authors.